# TDP-43 Proteinopathy Causes Broad Metabolic Alterations including TCA Cycle Intermediates and Dopamine Levels in Drosophila Models of ALS

**DOI:** 10.3390/metabo12020101

**Published:** 2022-01-21

**Authors:** Suvithanandhini Loganathan, Bryce A. Wilson, Sara B. Carey, Ernesto Manzo, Archi Joardar, Berrak Ugur, Daniela C. Zarnescu

**Affiliations:** 1Department of Molecular and Cellular Biology, University of Arizona, Tucson, AZ 85721, USA; suvitha@email.arizona.edu (S.L.); bryceawilson@email.arizona.edu (B.A.W.); sbcarey@email.arizona.edu (S.B.C.); manzoe@ohsu.edu (E.M.); ajoardar@gmail.com (A.J.); 2Departments of Neuroscience and Cell Biology, Yale University School of Medicine, New Haven, CT 06510, USA; berrak.ugur@yale.edu; 3Department of Neuroscience, University of Arizona, Tucson, AZ 85721, USA

**Keywords:** Amyotrophic lateral sclerosis (ALS), metabolic alterations, Tricarboxylic acid (TCA) cycle, glutamine, pyruvate, dopamine, pramipexole

## Abstract

Amyotrophic lateral sclerosis (ALS) is a fatal, complex neurodegenerative disorder that causes selective degeneration of motor neurons. ALS patients exhibit symptoms consistent with altered cellular energetics such as hypermetabolism, weight loss, dyslipidemia, insulin resistance, and altered glucose tolerance. Although evidence supports metabolic changes in ALS patients, metabolic alterations at a cellular level remain poorly understood. Here, we used a *Drosophila* model of ALS based on TDP-43 expression in motor neurons that recapitulates hallmark features of motor neuron disease including TDP-43 aggregation, locomotor dysfunction, and reduced lifespan. To gain insights into metabolic changes caused by TDP-43, we performed global metabolomic profiling in larvae expressing TDP-43 (WT or ALS associated mutant variant, G298S) and identified significant alterations in several metabolic pathways. Here, we report alterations in multiple metabolic pathways and highlight upregulation of Tricarboxylic acid (TCA) cycle metabolites and defects in neurotransmitter levels. We also show that modulating TCA cycle flux either genetically or by dietary intervention mitigates TDP-43-dependent locomotor defects. In addition, dopamine levels are significantly reduced in the context of TDP-43^G298S^, and we find that treatment with pramipexole, a dopamine agonist, improves locomotor function in vivo in *Drosophila* models of TDP-43 proteinopathy.

## 1. Introduction

Amyotrophic lateral sclerosis (ALS) is a progressive neurodegenerative disorder that affects both upper and lower motor neurons (MNs). It leads to the loss of motor coordination, paralysis, respiratory insufficiency, and eventual death within 30 months of symptom onset in most patients. Although several risk factors have been identified, the disease etiology remains poorly understood [1,2]. 

Several clinical studies have reported that ALS patients exhibit hypermetabolism, altered energy and lipid metabolism, weight loss, dyslipidemia, insulin resistance, and altered glucose tolerance, as reviewed in [3,4]. Metabolic interventions such as high calorie diets and metabolic supplements have been shown to delay disease progression in animal models, as reviewed in [4]. However, it remains unclear how these metabolic changes contribute to disease progression or selective degeneration of MNs. In addition, high body mass index and diabetes have been associated with increased survival in ALS patients [5]. Together, these studies highlight the need for a better understanding of the metabolic alterations associated with degenerating neurons to ultimately uncover potential therapeutic strategies for ALS.

One of the pathological hallmarks of ALS is the accumulation of TAR DNA binding protein (TDP-43) into irreversible cytoplasmic aggregates [6]. These cytoplasmic aggregates have been found in 97% of ALS cases irrespective of etiology [7]. To elucidate the molecular mechanisms underlying TDP-43 proteinopathies, we developed a *Drosophila* model of ALS based on the over-expression of TDP-43 in MNs. Notably, these models recapitulate several features of motor neuron (MN) disease including locomotor dysfunction, TDP-43 containing cytoplasmic aggregates, and reduced lifespan [8,9]. Additionally, similar to metabolic changes identified in patient tissues, *Drosophila* expressing TDP-43 in MNs exhibit alterations in glycolysis, the carnitine shuttle, and lipid beta oxidation [10,11,12,13]. Here, we show that TDP-43 proteinopathy causes additional, broad ranging metabolic alterations and highlight specific defects in TCA cycle intermediates and neurotransmitter levels. We find that manipulating the TCA cycle either genetically, through overexpression of *Drosophila* or human mitochondrial isocitrate dehydrogenase 3 (IDH3), or by feeding pyruvate, mitigates TDP-43 dependent locomotor defects. In addition, levels of dopamine are reduced, consistent with reports in ALS patients. We find that the administration of pramipexole, a dopamine agonist, improves locomotor function in *Drosophila* models of TDP-43 proteinopathy. These results show that dietary intervention or pharmacological and genetic manipulations mitigate TDP-43 induced locomotor dysfunction in vivo and identify broad ranging metabolic deficits that may inform therapeutic strategies based on improving cellular energetics.

## 2. Results

### 2.1. TDP-43 Proteinopathy in Motor Neurons Alters Several Metabolic Pathways In Vivo, in Drosophila Models of ALS

To study global metabolic alterations in ALS, we used *Drosophila* models of TDP-43 proteinopathy based on overexpression of wildtype TDP-43 (TDP-43^WT^) or an ALS-associated mutant variant, G298S (TDP-43^G298S^), in motor neurons, specifically. Metabolomic profiling was conducted on whole third instar larvae over-expressing TDP-43 in MNs (D42 > TDP-43^WT^ or D42 > TDP-43^G298S^) and genetic background controls (D42 > w^1118^) (see Materials and Methods for details). In brief, five replicates of 100 larvae each were submitted to Metabolon, Inc where they were analyzed for the presence of 572 compounds of known identity (metabolites). Biochemical profiling identified 106 significantly altered metabolites in TDP-43^WT^ compared to controls (32 upregulated and 74 downregulated, *p*-value < 0.05) and 217 significantly altered metabolites in TDP-43^G298S^ compared to controls (128 upregulated and 89 downregulated, *p*-value < 0.05) (see Appendix A). These analyses revealed alterations in several metabolic pathways including glycolysis and lipid beta oxidation that we have previously reported [12,13], as well as changes in neurotransmitter levels and TCA cycle intermediates, among others (e.g., amino acids, peptides, lipids, purine metabolism—see Appendix A).

### 2.2. TDP-43 Proteinopathy Causes Alterations in the TCA Cycle and the Glutaminolysis Pathway

Biochemical profiling showed that larvae expressing TDP-43^G298S^ exhibit a significant increase in several TCA cycle intermediates compared to controls (fumarate, 2.12-fold change, *p*-value = 0.032; malate, 2.22-fold change, *p*-value = 0.003; citrate, 3.6-fold change, *p*-value = 9.94 × 10^−6^; 2-methylcitrate, 1.7-fold change, *p*-value = 0.0005; aconitate, 1.6-fold change, *p*-value = 0.025; see Figure 1). In contrast, larvae expressing TDP-43^WT^ only showed a significant increase in fumarate (1.93-fold change, *p*-value = 0.036; see Figure 1). These results suggest at least a partial increase in TCA cycle flux in ALS.

The starting point of the TCA cycle is acetyl-CoA, which is derived from pyruvate. Our previously reported findings of significantly higher pyruvate levels in TDP-43 proteinopathy [13] support an increase in TCA cycle. However, not all TCA cycle intermediates are significantly increased in our models, either due to limited sensitivity or because additional input is required into the TCA cycle. Indeed, additional mechanisms exist, such as glutaminolysis, a pathway that replenishes TCA cycle intermediates by converting glutamine into glutamate, which, in turn, is converted into TCA cycle metabolites [14] (see Figure 1). Our metabolomic analyses indicate that glutamine and its acetylated analog, N-acetyl glutamine, are significantly increased in larvae expressing TDP43^G298S^ (glutamine, 1.13-fold change, *p*-value = 0.009; N-acetyl glutamine, 1.36-fold change, *p*-value = 0.007; see Figure 1), whereas N-acetyl glutamine and pyroglutamine were significantly increased in larvae expressing TDP-43^WT^ (N-acetyl glutamine, 1.26-fold change, *p*-value = 0.035; pyroglutamine, 1.21-fold change, *p*-value = 0.021; see Figure 1). These findings suggest that additional input into the TCA cycle via glutaminolysis is altered and provide a possible explanation for why not all metabolic intermediates (e.g., α-ketoglutarate) are significantly increased. Interestingly, upregulation of TCA cycle intermediates was also reported in patients [10]. Taken together, these data are consistent with TDP-43 dependent alterations in cellular energetics and suggest at least a partial increase in TCA cycle flux.

### 2.3. Overexpression of Isocitrate Dehydrogenase 3 (IDH3) in MNs Rescues TDP-43 Induced Locomotor Deficits

Our metabolomics data show that TCA cycle intermediates are upregulated in ALS; however, it remains unclear whether this is a direct consequence of TDP-43 proteinopathy or a compensatory mechanism. To address this issue and to further understand the role of TCA cycle in ALS pathology, we co-overexpressed mitochondrial isocitrate dehydrogenase 3 (IDH3) subunit a, the catalytic domain of IDH3 (human, hIDH3a or *Drosophila*, dIDH3a) [15], with TDP-43 in MNs and tested its effect on TDP-43-dependent locomotor defects using larval turning assays. IDH3a was chosen because it catalyzes an irreversible rate limiting step in the TCA cycle, namely oxidative decarboxylation of isocitrate. As previously shown [8,9], expression of TDP-43^WT^ or TDP-43^G298S^ significantly increased the larval turning times (10.38 ± 0.06 s, *p*-value < 0.0001 for TDP-43^WT^; 13.21 ± 0.07 s, *p*-value < 0.0001 for TDP-43^G298S^) compared to w^1118^ controls (8.19 ± 0.09 s). Interestingly, overexpression of either hIDH3a or dIDH3a in motor neurons on their own caused a significant increase in larval turning time (9.11 ± 0.03 0.06 s, *p*-value = 0.0134 for hIDH3a; 9.18 ± 0.03 s, *p*-value = 0.0165 for dIDH3a) (see Figure 2). However, when hIDH3a was co-overexpressed with TDP-43^WT^ or TDP-43^G298S^, it rescued TDP-43-induced locomotor deficits (7.97 ± 0.02 s, *p*-value < 0.0001 for TDP-43^WT^; 9.84 ± 0.06 s, *p*-value < 0.0001 or TDP-43^G298S^; see Figure 2A). Similarly, dIDH3a co-overexpressed with TDP-43^WT^ or TDP-43^G298S^ rescued TDP-43 induced locomotor deficits (7.97 ± 0.04 s, *p*-value < 0.0001 for TDP-43^WT^; 7.97 ± 0.02 s, *p*-value = 0.005 for TDP-43^G298S^; see Figure 2B). These findings show that increased IDH3a activity mitigates TDP-43-dependent locomotor deficits and suggest that the upregulation of TCA cycle intermediates is a compensatory mechanism in motor neurons.

### 2.4. Pyruvate Supplementation Mitigates TDP-43 Induced Locomotor Deficits in Drosophila Models of ALS

We have previously shown that the end product of glycolysis, pyruvate, is upregulated in TDP-43-expressing larvae, and genetic interaction experiments with phosphofructokinase (pfk, the rate limiting enzyme in glycolysis) showed that this is a compensatory mechanism (see Figure 3A and ref. [13]). This led us to hypothesize that pyruvate might have a neuroprotective effect in the context of TDP-43 proteinopathy. To investigate this possibility, we fed TDP-43 expressing larvae with pyruvate supplemented food and evaluated their locomotor function using larval turning assays [8,9]. These experiments showed that 250 μM sodium pyruvate caused a significant reduction in larval turning times in larvae expressing TDP-43^WT^ (12.46 ± 0.71 s, *p*-value = 0.035; see Figure 3B) or TDP-43^G298S^ (13.81 ± 0.54 s; *p*-value = 0.006; see Figure 3B) compared to larvae fed regular food (16.62 ± 0.91 s for TDP-43^WT^ and 18.75 ± 0.99 s for TDP-43^G298S^). These results indicate that high levels of pyruvate rescue TDP-43-induced locomotor deficits and are consistent with increased flux through the TCA cycle as a compensatory mechanism.

### 2.5. TDP-43 Proteinopathy Causes Alterations in Neurotransmitter Levels

Our metabolomics profiling data show altered neurotransmitter levels in larvae expressing TDP-43 consistent with altered neurotransmission. Specifically, biochemical profiling showed that larvae expressing TDP-43^G298S^ exhibit a significant reduction in dopamine, its metabolite 3,4-dihydroxyphenylacetate, and the dopamine precursor, l-DOPA, which has the capacity to cross the blood–brain barrier (dopamine, 0.28-fold change, *p*-value = 0.037; 3,4-dihydroxyphenylacetate, 0.43-fold change, *p*-value = 0.01; l-DOPA, 0.28-fold change, *p*-value = 0.037; see Figure 4). In larvae expressing TDP-43^WT^, we found a significant decrease in l-DOPA and 3,4-dihydroxyphenylacetate (l-DOPA, 0.45-fold change, *p*-value = 0.03; 3,4-dihydroxyphenylacetate, 0.51-fold change, *p*-value = 0.29; see Figure 4A) with no changes in dopamine levels. Notably, a reduction in the number of dopaminergic neurons has been seen in human ALS patients [16,17]. These changes in l-DOPA were accompanied by a significant decrease in its precursor, tyrosine (0.48-fold change, *p*-value = 0.048), in TDP-43^WT^, and a trend towards a decrease in TDP-43^G298S^ larvae (0.59-fold change, *p*-value = 0.064; see Appendix A).

Additional changes were detected in neurotransmitter precursors such as glutamine, aspartate, and putrescine, which were found to be significantly increased in larvae expressing TDP-43^G298S^ (glutamine, 1.13-fold change, *p*-value = 0.009; aspartate, 1.32-fold change, *p*-value = 0.005; putrescine, 1.3-fold change, *p*-value = 0.03; see Figure 4A). Although no significant changes were detected in glutamate or gamma-aminobutyrate (GABA), higher levels of the neurotransmitter precursors glutamine, aspartate, and putrescine in TDP-43^G298S^ could indicate decreased demand for neurotransmitter production. Interestingly, no significant difference was observed in larvae expressing TDP-43^WT^, possibly because the neurotransmission deficits caused by TDP-43^WT^ are not severe enough to alter precursor levels. Taken together, these data suggest that TDP-43 proteinopathy causes specific alterations in neurotransmission.

### 2.6. Dopamine Agonist Feeding Mitigates TDP-43 Induced Locomotor Deficits in Drosophila Models of ALS

Our data showing that dopamine levels are decreased in the context of mutant TDP-43 overexpression led us to hypothesize that pramipexole, a dopamine agonist that has been shown to reverse motor deficits caused by dopamine depletion in Parkinson’s disease patients, may also be protective in the context of TDP-43 proteinopathy. To investigate if pramipexole can rescue locomotor deficits in *Drosophila*, we fed TDP-43^G298S^-expressing larvae with either 5 or 10 μM pramipexole-supplemented food and evaluated their locomotor function. These experiments showed that, while 5 μM pramipexole had no effect (see Appendix A), 10 μM pramipexole caused a significant reduction in larval turning times in larvae expressing TDP-43^G298S^ (17.15 ± 1.31 s; *p*-value = 0.03; see Figure 4B) compared to larvae fed regular food (21.52 ± 0.9 s for TDP-43^G298S^). These results indicate that a dopamine agonist rescues TDP-43 induced locomotor deficits in vivo.

## 3. Discussion

Clinical observations identified metabolic alterations in ALS patients that are consistent with altered cellular energetics reviewed in [3]. However, the relationship between whole body and cellular level alterations remains poorly understood. To uncover TDP-43-dependent changes in neuronal metabolism, we determined the metabolome of *Drosophila* ALS models based on the overexpression of TDP-43 in MNs. Notably, our model has previously identified metabolic and molecular alterations that were subsequently validated in ALS spinal cords and induced pluripotent stem cell (iPSC)-derived MNs [13,18,19]. Since metabolic profiling was performed with whole larvae while TDP-43 was overexpressed in MNs only, we anticipate that some of the biochemical alterations we detected may be caused by cell non-autonomous effects (e.g., glia), and that some TDP-43 proteinopathy-dependent changes were missed due to a “dilution effect”.

In this study, we used global metabolomics to gain insight into altered metabolic pathways in ALS, which could uncover potential therapeutic strategies. Here we report several altered metabolites including amino acids, peptides, lipids, and purine metabolism (see Appendix A), several of which have also been found to be changed in ALS patients [10,11,20]. We identified a significant increase in several TCA cycle intermediates, which is consistent with increased TCA flux. Interestingly, alterations in the TCA cycle have also been observed in ALS patients [10] and in an MN-like cell line, NSC-34 model [21]. We also identified increased levels of glutamine but no changes in glutamate, one of its breakdown products. This is in contrast with reports of increased glutamate levels in the blood and cerebrospinal fluid of ALS patients [22,23,24] and may be attributed to differences between *Drosophila* and human metabolism and/or due to limitations in our experimental approach (i.e., the “dilution effect”). Alternatively, it is possible that our *Drosophila* model of ALS does not recapitulate all the aspects of motor neuron disease in humans.

To further understand the contribution of the TCA cycle defects to TDP-43-dependent toxicity, we used genetic and dietary interventions. IDH3a catalyzes the production of α-ketoglutarate, which can also be replenished by glutaminolysis. In the absence of glutaminolysis, we tested whether IDH3a overexpression in the context of TDP-43 can rescue TDP-43-induced locomotor deficits. Indeed, we found that the overexpression of either human or *Drosophila* IDH3a, the catalytic subunit of NAD+-dependent mitochondrial isocitrate dehydrogenase, although toxic on their own, mitigate locomotor deficits in the context of TDP-43. Similarly, dietary supplementation with pyruvate, which is predicted to fuel the TCA cycle, also mitigates TDP-43-induced locomotor deficits. Alternatively, pyruvate could be taken up into glia, where it is converted into lactate and then taken back up into neurons for fueling ATP production (i.e., the glia–neuron lactate shuttle) [25]. These results are consistent with previous findings that Triheptanoin slows motor neuron loss in a mouse model of ALS by replenishing TCA cycle intermediates [26]. TCA cycle is critical for both energy production and biosynthesis [27]. While the precise mechanism remains unknown, our data suggest that upregulating TCA cycle might be neuroprotective in ALS.

At this time, we cannot distinguish whether pyruvate supplementation is protective due to increasing the TCA cycle or due to acting as free radical scavenger, as reported in tumor cells [28]. It will also be interesting to test whether supplementation with α-keto glutarate is protective, as predicted by the rescue effect of IDH3a overexpression. Interestingly, IDH3a was recently shown to play a role in coupling mitochondrial metabolism to synaptic function [15]. We speculate that IDH3a overexpression in the context of TDP-43 proteinopathy may improve neuromuscular junction physiology, which we have previously shown to be altered in *Drosophila* expressing TDP-43 in MNs [19] (see Figure 5 for model).

Our findings that dopamine levels are reduced are consistent with previous reports that dopaminergic neurons are reduced in a subset of ALS patients [16,17]. It remains to be determined whether pramipexole administration improves locomotor function by enhancing mitochondrial metabolism (as predicted for dexpramipexole, its R(+) enantiomer) or by enhancing dopaminergic signaling (see Figure 5 for model). We acknowledge that dexpramipexole, although promising in Phase II studies [29], subsequently failed in Phase III clinical trials (Biogen Idec). However, recently, new dopamine agonists have shown promise in ALS patient-derived iPSCs [30,31], suggesting that this strategy may be worth reevaluating.

Taken together, our global metabolomic study identifies broad-ranging metabolite alterations affecting several pathways including glycolysis, the carnitine shuttle, and lipid beta oxidation, which we have previously reported [12,13], as well as the TCA cycle and neurotransmitter systems, which we report in this manuscript. These findings suggest that cellular metabolism provides novel opportunities for devising therapeutic strategies based on restoring metabolic homeostasis in ALS and related neurodegenerative disorders.

## 4. Materials and Methods

### 4.1. Drosophila Genetics 

D42-GAL4 was used to drive transgene expression in motor neurons. D42-GAL4 virgin female flies were crossed with w^1118^, UAS-TDP43^WT^-YFP, UAS-TDP43^G29S8^-YFP described in [8,9], UAS-hIDH3a, or UAS-fIDH3a described in [15] males. All fly lines were maintained on molasses cornmeal media, at 25 °C with a 12 h dark/light cycle.

### 4.2. Fly Food Supplementation 

Standard yeast/cornmeal/molasses food was heated and allowed to cool to ~55 °C. For pyruvate supplementation, sodium pyruvate (Gibco; Life Technologies Europe B.V., Bleiswijk, The Netherlands; Catalog No. 11360070) was added to reach a final concentration of 250 μM. The food was then dispensed into vials and allowed to solidify. For pramipexole supplementation, pramipexole (Selleckchem, Houston, TX, USA; Catalog No. S2460) was dissolved in DMSO and was added to the food to reach a final concentration of 10 μM.

### 4.3. Metabolomics 

Metabolite analyses were performed by Metabolon. Inc as previously described [12,13,32]. First, 50–60 wandering third instar larvae (50–60 mg) per sample were collected and flash-frozen in liquid nitrogen. Five replicates were collected per genotype. The samples were processed using ultrahigh performance liquid chromatography/mass spectrometry (UHPLC/MS), or by gas chromatography/mass spectrometry (GC/MS). One-way ANOVA was used to identify biochemicals that significantly altered between experimental genotypes.

### 4.4. Locomotor Assays 

Larval turning assays were performed as previously described [8,9]. Briefly, third instar larvae were placed on a grape juice agar surface and were allowed to acclimate for 30 s. Then, the larvae were turned ventral side up using a paint brush. The time taken for the larvae to turn ventral side down and make a full forward motion was measured. At least 30 larvae were assayed per genotype.

### 4.5. Statistics 

Larval turning data were analyzed using Kruskal–Wallis tests using GraphPad Prism v9.0.

## Figures and Tables

**Figure 1 metabolites-12-00101-f001:**
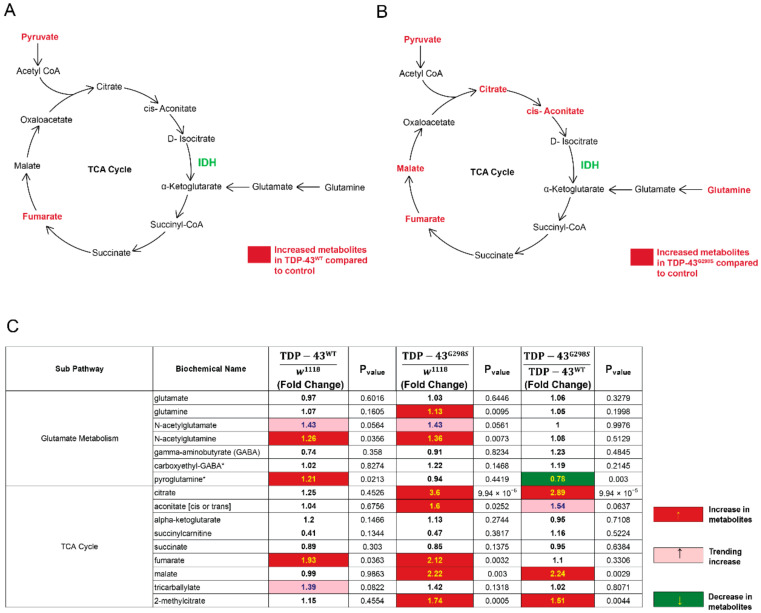
Metabolomic profiling uncovers alterations in TCA cycle and glutamate metabolism. (**A**) TCA cycle and glutamate metabolism components altered in larvae expressing TDP-43^WT^. (**B**) TCA cycle and glutamate metabolism components altered in larvae expressing TDP-43^G298S^. (**C**) Summary of metabolite fold changes in TDP^WT^ and TDP^G298S^ compared to w^1118^ controls, and TDP^G298S^ compared to TDP^WT^. Red indicates significant upregulation; pink indicates an upward trend; green indicates significant downregulation. Statistical significance was determined using one-way ANOVA. * indicates compounds that have not been officially confirmed based on a standard, but we are confident in its identity.

**Figure 2 metabolites-12-00101-f002:**
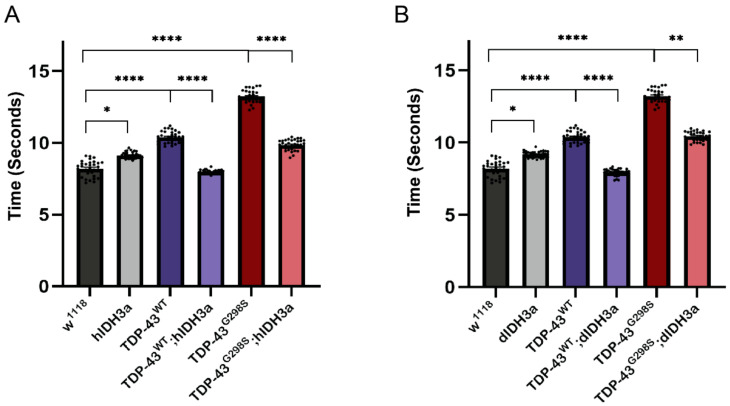
Overexpression of IDH3a rescues TDP-43 induced locomotor defects. (**A**) Larval turning times of larvae expressing human IDH3a (hIDH3a), TDP-43^WT^, or TDP-43^G298S^, individually or together, as indicated. (**B**) Larval turning times of larvae expressing *Drosophila* IDH3a (dIDH3a), TDP-43^WT^, or TDP-43^G298S^, individually or together, as indicated. *n* = 30 larvae per genotype. Kruskal–Wallis was used to determine statistical significance. *—*p*-value < 0.05, **—*p*-value < 0.01, ****—*p*-value < 0.0001.

**Figure 3 metabolites-12-00101-f003:**
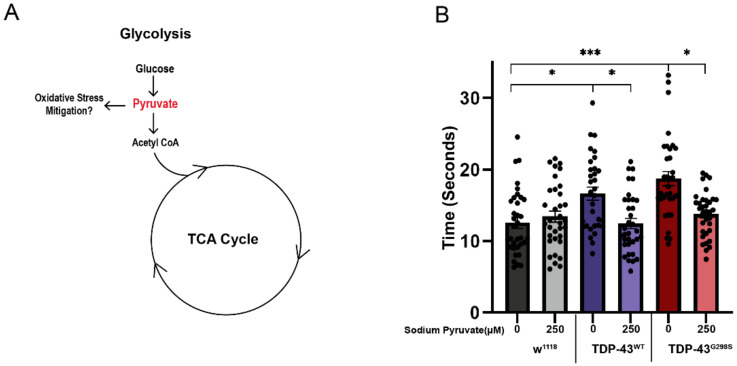
Dietary supplementation of pyruvate rescues TDP-43 induced locomotor defects. (**A**) A flowchart showing potential pyruvate input into multiple metabolic pathways. (**B**) Larval turning assays for *Drosophila* expressing TDP-43 fed a cornmeal-based media supplemented with 250 μM sodium pyruvate, as indicated on the *x*-axis. Genotypes and treatments are as indicated. *n* = 30 larvae per treatment/per genotype. Kruskal–Wallis test was used to determine statistical significance. *—*p*-value < 0.05, ***—*p*-value < 0.001.

**Figure 4 metabolites-12-00101-f004:**
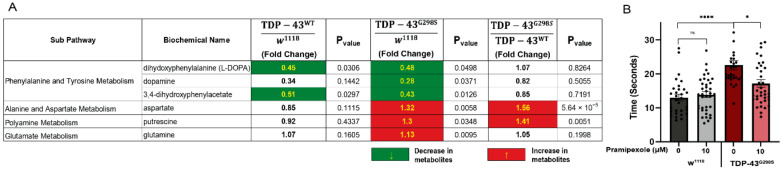
TDP-43 proteinopathy induces alterations in neurotransmitters and their precursors. (**A**) Summary of significantly altered neurotransmitters, their metabolites, and precursors in TDP^WT^ and TDP^G298S^ compared to w^1118^ controls. Red indicates significant upregulation and green indicates significant downregulation. Statistical significance was determined using one-way ANOVA. (**B**) Larval turning assays for *Drosophila* expressing TDP-43^G298S^ fed a cornmeal-based media supplemented with 10 μM pramipexole, as indicated on the *x*-axis. DMSO was used as vehicle. *n* = 30 larvae per treatment/per genotype. Kruskal–Wallis test was used to determine statistical significance. *—*p*-value < 0.05, ****—*p*-value < 0.0001.

**Figure 5 metabolites-12-00101-f005:**
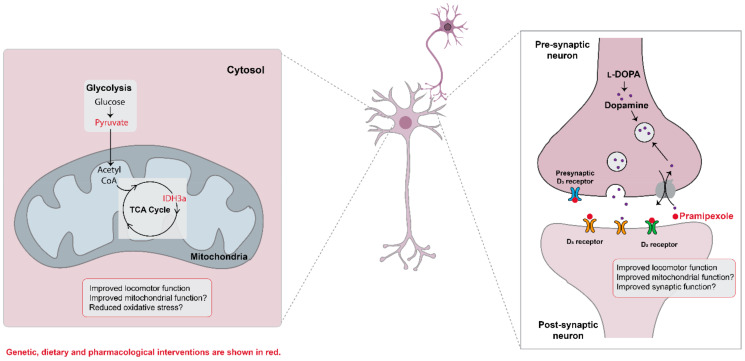
Proposed model depicting metabolic pathways altered in TDP-43 proteinopathy. Among several metabolic changes, the TCA cycle is upregulated, likely as a compensatory mechanism. Pyruvate treatment or IDH3a overexpression improves locomotor function, either by improving mitochondrial function and/or by mitigating oxidative stress. Additionally, neurotransmitter systems are altered, including dopamine. Pramipexole, a dopamine agonist, mitigates TDP-43 induced locomotor deficits, either by improving mitochondrial and/or synaptic function. Dietary, genetic, or pharmacological interventions are shown in red.

## Data Availability

Data is contained within the article or Appendix A.

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
