# Peer review of "TDP-43 Proteinopathy Causes Broad Metabolic Alterations including TCA Cycle Intermediates and Dopamine Levels in Drosophila Models of ALS"

_metabolites, 2022, doi:10.3390/metabo12020101_

Round 1

Reviewer 1 Report

Summary

Loganathan et al present a manuscript using a Drosophila model of ALS which involves the expression of TDP-43 (WT and mutant), the model has previously been used and is an accepted model of motor neuron disease in ALS. Using an unbiased metabolomics approach, they were able to uncover changes in amino acids and other metabolites in flies upon TDP-43 overexpression. These changes include reduced neurotransmitter (dopamine) and increases neurotransmitter precursors (putrescine, aspartate, glutamine) which suggests reduced demand for neurotransmitters. Indeed, the authors can demonstrate through rescue experiments that supplementation with pyruvate or treatment with a dopamine agonist can improve the motor phenotypes observed in the mutant flies which support the reduced neurotransmitter point.

Brief Comments

Generally, the experiments are well controlled, and the study is presented in a clear and reproducible manner. The central points of the manuscript do build on previous research done and at times if the results presented are from previous articles or derived from this one can be confusing, as pointed out in the specific comments below. With minor changes, additions, or clarifications the manuscript would be greatly improved.

Specific Comments

Some areas that can specifically be improved by the authors:

  • Lines 53-64 outline the scope and findings of the work, however, this takes up a significant portion of the introduction. Consider shortening this as it then becomes repetitive in the results.
  • Line 82- the authors make reference to the Supplemental Materials, this was not provided with the review package. Indeed, a more complete and thorough review could be completed if these missing details were provided.
  • At line 90 the authors correctly suggest it may be due to increased flux. Here it is important to note that the change in 4 carbon TCA intermediates is an excellent gauge of increased flux, therefore it would be worthwhile to see the levels of other 4 carbon molecules such as malate, succinate and OAA, to better gauge if they support this logical contention.
  • At lines 130, the authors stating references 8 and 9 at the end of the sentence could be confused with that data being presented in the sentence. Perhaps a better placement for the intext citation would be at the end of the phrase as “previously shown”.
  • The sentence starting at line 151 needs a reference, furthermore, more clarity is needed on this statement about what exactly is increased in glycolysis. The next sentence does speak about pyruvate but perhaps a reference or restructuring would create a better linking for the reader to appreciate the point being made.
  • Line 184, the point of neurotransmitter precursors being altered is made very well here. So given the decreases in DOPA and dopamine levels, it would be interesting to see what levels of tyrosine looked like. It is possible that this may be in the supplemental but since this is not attached it is not possible to discern if they are elevated or not.
  • At line 204, what is the rationale for specifically 10 uM Pramipexole, is this established from literature, in which case this should be indicated with a reference.
  • Observing the data between Fig 2 & Fig 3 compared to Fig 4, the time for larval turning in the mutant is higher when treated with the vehicle control in Fig 4. This point does not seem to be noted by the authors and a brief statement on this may be useful for the reader to take these results in context. If there is information about this interaction (DMSO increasing larval turning time) being seen in the literature, it should be referenced.
  • At line 240, the author state two possibilities but a third presents itself, that is that the TDP-43 expression model does not capture all features of ALS phenotypically as it relates to motor neuron disease. Perhaps the authors allude to this in the statement on differences in the model, but this point should perhaps be made more explicitly.

Minor points

  • The title is lengthy and difficult to follow as a result, the authors may wish to consider reducing the length of the title to be more succinct while still capturing the breadth of the work
  • At line 38 there is a PMID reference, this should be adjusted to the standard referencing format in time for publication
  • At like 252, the authors point out the glia-lactate shuttle but do not provide a suitable reference for this that would assist the reader.
  •  

Figures & tables

  • Figure 1C- it is curious why the heading Fold change is not used in the table, the formulae can remain but at least the title may more quickly help to acquaint the reader with what parameter is being shown.
  • Figure 2- the use of the semicolon in the x-axis labels is confusing, would a ‘+’ not more accurately represent this condition since the human or drosophila IDH3a would be an exogenous expression?
  • Figure 3A- proposes oxidative stress mitigation but then in Figure 5, this is only referred to by text at the bottom. Some members of the TCA cycle would contribute to OS mitigation and perhaps this should be indicated in Figure 5 with the IDH3a then being shown as an enzyme that can lead to this. If the authors think this is too speculative it can be left out. But there appears to be a disconnect between these two figures.
  • Figure 3B- the legend should be clarified to indicate that only 250 uM was used not “concentrations indicated” since only one is used.
  • Figure 4- the title should be adjusted to include precursors, as neurotransmitters alone are not presented in this figure. In fact, most molecules are precursors in that table.
  • Figure 5- perhaps identifying what the text in red indicates visually on the figure would assist the reader and make a glance of the figure independent of the legend.

Author Response

We thank the reviewers for their thoughtful comments and suggestions. We have addressed their concerns below, point by point, and hope that the manuscript is now acceptable for publication.

Reviewer 2 Report

The manuscript from Loganathan et al. is focused on the analysis of metabolic alterations caused by TDP-43 (wild-type or mutant) overexpression in Drosophila larvae. The current study, a follow up on a previous manuscript detailing the compensatory upregulation of glycolysis in ALS, identified increased levels in several metabolites of the TCA cycle, as well as altered levels of few neurotransmitters, including dopamine. The authors go on to prove that genetic or pharmacologic modulation of these two main pathways can improve motor behavior in TDP-43 overexpressing fly larvae. The study is well designed and well controlled, and the data is elegantly presented. Metabolic alterations are emerging as key changes occurring during the degenerative process in ALS and other neurodegenerative diseases, and this study adds important novel information to our understanding of the role of mitochondria energy metabolism in TDP-43 linked ALS.
A few minor comments are listed below.
1.    The metabolomic data presented were analyzed by comparing w1118 to TDP-43 WT flies or w1118 to TDP-43 G298S flies. While both wild type and mutant TDP-43 overexpression are known to cause neurodegeneration in human and animal models of ALS, the authors could consider including a comparison between WT and G298S flies to investigate mutation-specific effects on fly metabolism.
2.    The authors show that manipulation of the TCA cycle rescues motor behavior in TDP-43 flies. However, the mechanism by which this occurs is only implied but not demonstrated. The authors could consider investigating if and how pyruvate supplementation or IDH3a overexpression alter the levels of TCA cycle metabolites. It would be also interesting to assess whether the effect of IDH3a and pyruvate supplementation is additive, synergistic, or neither. 
3.    Larval turning times shown in Figure 2 and Figure 3 are not consistent. 
4.    While glutaminolysis is an important input into the TCA cycle via α-ketoglutarate, it is similarly relevant that α-ketoglutarate can be shunted from the TCA cycle for the synthesis of glutamate. Since glutamine levels are increased in mutant TDP-43 flies, is it possible that that is caused by increased in the α-ketoglutarate -> glutamate -> glutamine pathway?
5.    Page 1, line 19 should read: “identified significant alterations “in” several metabolic pathways 
6.    Page 4, line 139: “locmotor” should be “locomotor”

Author Response

(The authors gave the same response as above.)

Reviewer 3 Report

In the current manuscript, authors used a fly (Drosophila) model of ALS based on TDP-43 expression in motor neurons that recapitulates hallmark features of motor neuron disease including TDP-43 aggregation, locomotor dysfunction and reduced lifespan. To understand further mechanism of TDP-43, they we performed global metabolomic profiling in larvae expressing TDP-43 (WT or ALS associated mutant variant, G298S) and identified significant alterations several metabolic pathways. They report alterations in multiple metabolic pathways and highlight upregulation of TCA cycle metabolites and defects in neurotransmitter levels.

They also show that modulating TCA cycle flux either genetically or by dietary intervention mitigates TDP-43 dependent locomotor defects. Dopamine levels are significantly reduced in the context of TDP-43G298S and treatment with pramipexole, a dopamine agonist, improves locomotor function in vivo, in flies. These observations are important and worth reporting.

My only suggestion is that authors need to comment how and why TCA cycle is impacted by TDP-43G298S, if yes, explain the activation of TCA cycle is a compensatory aspect ALS and imply in humans with ALS.

Author Response

(The authors gave the same response as above.)

Round 2

Reviewer 1 Report

The authors have done a good job of revising the manuscript and preparing it for publication. The authors should be congratulated on their work.

Author Response

Thank you, for the suggestions to improve our manuscript.